# Immortalized Canine Adipose-Derived Mesenchymal Stem Cells as a Novel Candidate Cell Source for Mesenchymal Stem Cell Therapy

**DOI:** 10.3390/ijms24032250

**Published:** 2023-01-23

**Authors:** Yuyo Yasumura, Takahiro Teshima, Tomokazu Nagashima, Takashi Takano, Masaki Michishita, Yoshiaki Taira, Ryohei Suzuki, Hirotaka Matsumoto

**Affiliations:** 1Laboratory of Veterinary Internal Medicine, Department of Veterinary Clinical Medicine, School of Veterinary Medicine, Faculty of Veterinary Science, Nippon Veterinary and Life Science University, 1-7-1 Kyonan-cho, Musashino, Tokyo 180-8602, Japan; 2Research Center for Animal Life Science, Nippon Veterinary and Life Science University, 1-7-1 Kyonan-cho, Musashino, Tokyo 180-8602, Japan; 3Laboratory of Veterinary Pathology, Department of Veterinary Clinical Medicine, School of Veterinary Medicine, Faculty of Veterinary Science, Nippon Veterinary and Life Science University, 1-7-1 Kyonan-cho, Musashino, Tokyo 180-8602, Japan; 4Laboratory of Veterinary Public Health, Department of Veterinary Clinical Medicine, School of Veterinary Medicine, Faculty of Veterinary Science, Nippon Veterinary and Life Science University, 1-7-1 Kyonan-cho, Musashino, Tokyo 180-8602, Japan

**Keywords:** cell cycle, cell senescence, CD90, cyclin-dependent kinase 4, dog, immortalization, mesenchymal stem cell, telomerase reverse transcriptase

## Abstract

Mesenchymal stem cells are expected to be a cell source for stem cell therapy of various diseases in veterinary medicine. However, donor-dependent cell heterogenicity has been a cause of inconsistent therapeutic efficiency. Therefore, we established immortalized cells from canine adipose tissue-derived mesenchymal stem cells (ADSCs) to minimize cellular heterogeneity by reducing the number of donors, evaluated their properties, and compared them to the primary cells with RNA-sequencing. Immortalized canine ADSCs were established by transduction with combinations of the R24C mutation of human cyclin-dependent kinase 4 (CDKR24C), canine cyclin D1, and canine TERT. The ADSCs transduced with CDK4R24C, cyclin D1, and TERT (ADSC-K4DT) or with CDK4R24C and cyclin D1 (ADSC-K4D) showed a dramatic increase in proliferation (population doubling level > 100) without cellular senescence compared to the primary ADSCs. The cell surface markers, except for CD90 of the ADSC-K4DT and ADSC-K4D cells, were similar to those of the primary ADSCs. The ADSC-K4DT and ADSC-K4D cells maintained their trilineage differentiation capacity and chromosome condition, and did not have a tumorigenic development. The ability to inhibit lymphocyte proliferation by the ADSC-K4D cells was enhanced compared with the primary ADSCs and ADSC-K4DT cells. The pathway analysis based on RNA-sequencing revealed changes in the pathways mainly related to the cell cycle and telomerase. The ADSC-K4DT and ADSC-K4D cells had decreased CD90 expression, but there were no obvious defects associated with the decreased CD90 expression in this study. Our results suggest that ADSC-K4DT and ADSC-K4D cells are a potential novel cell source for mesenchymal stem cell therapy.

## 1. Introduction

Mesenchymal stem cells (MSCs) can be isolated from many different tissues, including bone marrow, adipose tissues, dental pulp, and umbilical cord [1,2,3]. Adipose tissue-derived MSCs (ADSCs) have been investigated extensively for the treatment of various diseases because of the minimal invasiveness of their harvesting and high cell-proliferative capacity compared with bone marrow-derived MSCs in veterinary medicine [4,5,6,7,8,9]. The clinically expected functions of ADSCs are anti-inflammatory and immunomodulatory effects through secreted factors, including exosomes, miRNAs, cytokines, chemokines, and growth factors [10,11,12,13,14]. However, the clinical efficacy in multiple trials of ADSCs has been variable with results that are not consistent with pre-clinical findings in vitro [15,16]. A reason for this may be the heterogeneity of ADSCs [17,18,19]. The source of ADSCs is a major variable affecting their therapeutic efficacy. Therefore, donor-to-donor variability of the phenotype and growth kinetics causes significant inter-individual heterogeneity in the secreted factors of ADSCs. This potentially results in inconsistent outcomes in clinical trials and prevents the practical application of stem cell therapies [16,19]. A solution to avoid variations in ADSCs and stabilize the therapeutic effects is to restrict donors, but this is impossible because of the limited proliferative capacity of primary ADSCs. Therefore, in this study, we attempted to immortalize canine ADSCs.

There are several approaches to establish immortalized cell lines [20]. The classical immortalization method is using viral vectors to introduce ectopic expression of telomerase reverse transcriptase (TERT), human papillomavirus (HPV)-E6/E7 oncogenes, and simian virus 40 (SV40) large T antigen. HPV-E6/E7 and SV40 T antigen bind to negative cell cycle regulators, such as p53 and the retinoblastoma gene product, which act as tumor suppressor proteins in the genome [21]. Moreover, enhanced telomerase activity is essential for the immortalization of cells [22]. Therefore, combinations of human TERT (hTERT) and HPV-E6/E7 or SV40 T antigen expression have been reported to efficiently immortalize human primary cells [23]. However, the overexpression of these oncogenic proteins raises concerns about inducing genomic instability and affecting the properties of the original primary cells. To overcome these concerns, a novel immortalization method was developed by expression of R24C mutant cyclin-dependent kinase 4 (CDK4R24C), cyclin D1 (CCND1), and hTERT [24]. This method efficiently establishes immortalized human myogenic cell lines that maintain their original phenotype. Moreover, apart from humans, immortalization can be effectively induced in cells of various species [25,26,27]. Therefore, in this study, we aimed to establish immortalized canine ADSCs transduced with combinations of human CDK4R24C, canine CCND1, and canine TERT, and analyze the biological characteristics of these cells including whether they maintained MSC properties.

## 2. Results

### 2.1. Enhanced Proliferation of ADSC-K4DT and ADSC-K4D Cells

We performed sequential passaging to determine the cell proliferation ability (Figure 1). Although the primary ADSCs had almost ceased proliferation at approximately population doubling level (PDL) 15, the ADSC-K4DT and ADSC-K4D cells continued cell proliferation to >PDL 100 during the 3 months of observation. The proliferation rate of the ADSC-K4D cells was higher than that of the ADSC-K4DT cells. However, similar to the primary ADSCs, the ADSC-TERT cells did not maintain proliferation and had almost ceased proliferation at PDL 15.

### 2.2. Cell Cycle of ADSC-K4D and ADSC-K4DT Cells

We carried out a cell cycle analysis at PDL 50 of the ADSC-K4DT and ADSC-K4D cells (Figure 2). The cell cycle duration was shortened in the ADSC-K4DT and ADSC-K4D cells compared with the primary ADSCs. The ratio of the ADSC-K4DT (49.6 ± 3.3%) and ADSC-K4D (48.7 ± 2.6%) cells in G0/G1 phase was significantly decreased compared with that of the primary ADSCs (72.9 ± 2.3%), and the ratio of the ADSC-K4DT (44.8 ± 3.5%) and ADSC-K4D (45.2 ± 3.8%) cells in S and G2/M phases was significantly increased compared with that of the primary ADSCs (24.6 ± 2.7%).

### 2.3. CD90 Expression Is Altered in ADSC-K4D and ADSC-K4DT Cells

The primary ADSCs expressed the MSC markers CD29, CD44, and CD90, and very few expressed CD34, CD45, and HLA-DR. Even after numerous cell divisions, the expression pattern of the cell surface markers, except for CD90 in the ADSC-K4DT and ADSC-K4D cells, was similar to the primary ADSCs (Figure 3). As shown in Figure 4 and Table 1, CD90 expression in the ADSC-K4D cells was consistently slightly lower, whereas that in the ADSC-K4DT cells was decreased markedly as division progressed compared with that in the primary ADSCs.

### 2.4. Maintenance of the Trilineage Differentiated Ability of ADSC-K4DT and ADSC-K4D Cells

We determined whether the ADSCs maintained their trilineage differentiation ability after transduction with canine TERT, canine CCND1, or human CDK4R24C. As shown in Figure 5, the ADSC-K4DT and ADSC-K4D cells were able to differentiate into adipocytes, osteocytes, and chondrocytes similarly to the primary ADSCs.

### 2.5. Lack of Cellular Senescence in ADSC-K4D and ADSC-K4DT Cells

The primary ADSCs and ADSC-TERT cells showed enlarged cytoplasm at approximately PDL 15, but the ADSC-K4DT and ADSC-K4D cells showed no morphological changes (Figure 6). Most of the primary ADSCs and ADSC-TERT cells that had almost ceased proliferation at approximately PDL 15 were positive for senescence-associated β-galactosidase (SA-β-gal). However, few cells that had undergone repeated cell division of approximately PDL 100 among the ADSC-K4DT and ADSC-K4D cells were positive for SA-β-gal (Figure 6).

### 2.6. TERT, CCND1, and CDK4R24C Expression in Primary ADSCs, and ADSC-K4DT and ADSC-K4D Cells

In the primary ADSCs, canine CCND1 expression was observed, but canine TERT was expressed even in the cells in proliferative periods. Amplification products specific to canine TERT, canine CCND1, and human CDK4R24C were detected in the ADSC-K4DT cells (Figure 7). Canine TERT expression was observed in the ADSC-K4D cells despite not being transduced with canine TERT.

### 2.7. Maintenance of the Chromosome Condition in ADSC-K4DT and ADSC-K4D Cells

We performed karyotype analysis of 50 mitotic primary ADSCs (passage 3), ADSC-K4DT cells (PDL 105), and ADSC-K4D cells (PDL 102). All cells in each cell line had 2 n = 78, indicating that the ADSC-K4DT and ADSC-K4D cells had maintained the original number of chromosomes (Figure 8). Next, we performed G-banding analysis of 20 mitotic cells, which allowed for the identification of individual chromosomes by the banding patterns. There were no chromosome abnormalities in the ADSC-K4DT or ADSC-K4D cells.

### 2.8. Non-Tumorigenicity of ADSC-K4DT and ADSC-K4D Cells

Tumor development was observed at sites injected with Hela cells (all 15 sites: Figure 9B), whereas no tumor formation sites were observed in the sites injected with primary ADSCs or ADSC-K4DT and ADSC-K4D cells in groups 1–3 after 30 days (Figure 9A). Furthermore, no tumors were generated at all sites injected with primary ADSCs or ADSC-K4DT and ADSC-K4D cells even after 16 weeks in groups 4 and 5 (Figure 9C).

### 2.9. ADSC-K4DT and ADSC-K4D Cells Inhibit PBMC Proliferation

The proliferation rate of concanavalin A (ConA)-stimulated peripheral blood mononuclear cells (PBMC) cocultured with primary ADSCs was significantly decreased compared with that of ConA-stimulated PBMCs (stimulated PBMCs: 83.1 ± 2.5%; cocultured with primary ADSCs: 71.3 ± 1.4%). The ADSC-K4DT cells inhibited the ConA-stimulated PBMCs similarly to the primary ADSCs, but the ADSC-K4D cells strongly inhibited the ConA-stimulated PBMCs compared with the primary ADSCs and ADSC-K4DT cells (Figure 10). The suppressive effect of the ADSC-K4DT and ADSC-K4D cells on the PBMCs remained as cell division progressed.

### 2.10. Differentially Expressed Genes among Primary ADSCs, and ADSC-K4DT and ADSC-K4D Cells

RNA-seq detected expression of 13,199 genes among the primary ADSCs, and the ADSC-K4DT and ADSC-K4D cells. The gene expression levels are shown in Appendix A. By comparing the primary ADSCs and ADSC-K4DT cells, there were 1418 genes with upregulated expression in the ADSC-K4DT cells and 1200 genes with downregulated expression in the ADSC-K4DT cells. By comparing the primary ADSCs and ADSC-K4D cells, there were 1653 genes with upregulated expression in the ADSC-K4D cells and 1967 genes with downregulated expression in the ADSC-K4D cells. By comparing the ADSC-K4DT and ADSC-K4D cells, there were 1331 genes with upregulated expression in the ADSC-K4DT cells and 585 genes with upregulated expression in the ADSC-K4D cells.

### 2.11. Differences in Pathways among Primary ADSCs, and ADSC-K4DT and ADSC-K4D Cells

Using differentially expressed genes, we carried out the pathway analysis between groups. The pathway analysis using RaNA-seq data was performed with the KEGG, REACTOME, and WikiPathways databases. By comparing the ADSC-K4DT and ADSC-K4D cells with the primary ADSCs, some pathways related to the cell cycle and telomeres were revealed (Appendix A). The cell cycle in the KEGG pathway database contains 133 genes, 113 of which were detected with RNA-seq. The expression levels of these 113 genes are presented as a heatmap, and the differentially expressed genes were mapped to the KEGG cell cycle pathway map (Figure 11). Cellular senescence in the KEGG pathway database contains 159 genes, 138 of which were detected with RNA-seq. The expression levels of these 138 genes are presented as a heatmap, and the differentially expressed genes were mapped to the KEGG cellular senescence pathway map (Figure 12).

## 3. Discussion

We attempted the immortalization of canine ADSCs by transduction with combinations of human CDKR24C, canine CCND1, and canine TERT genes. After repeated passaging, TERT transduction alone did not immortalize the canine ADSCs, whereas K4DT or K4D transduction successfully immortalized the cells. In a previous report, rat ADSCs were successfully immortalized by transduction with hTERT alone [28], whereas immortalization was not achieved with hTERT alone in human ADSCs but rather by transduction in combination with either SV40 or HPV-E6/E7 [29]. Flow cytometry confirmed that canine TERT was transduced into >90% of the primary ADSCs, and integration of canine TERT into the ADSCs was confirmed with PCR, but cell proliferation was arrested at approximately PDL 15. Conversely, the ADSC-K4DT and ADSC-K4D cells maintained proliferation for PDL >100. In a report of immortalization of human dental pulp stem cells, K4DT cells did not show decreased proliferation, whereas K4D cells had a slower proliferation rate at approximately passage 4 [30]. Additionally, senescence of the K4D cells was indicated by an enlarged cytoplasm, and the cells stained positively for SA-β-gal. However, in the immortalization of Tsushima leopard cat fibroblasts, proliferative capacity was maintained by K4D and K4DT cells [25]. The results of enzymatic activity of telomere elongation led to the conclusion that the combination of K4DT transduction was an effective immortalization method. In our study, the proliferation of the ADSC-K4D cells was faster than that of the ADSC-K4DT cells, and the ADSC-K4D and ADSC-K4DT cells did not show cellular senescence with SA-β-gal staining. Furthermore, canine TERT expression was detected in the ADSC-K4D cells that were not transduced with canine TERT, and the expression levels of canine TERT determined with RNA-seq were similar between the ADSC-K4D and ADSC-K4DT cells.

Several combinations of transduced genes have been reported for ADSC immortalization, namely hTERT and SV40 T antigen, hTERT and HPV-E6/E7, and murine Bmi-1 and hTERT [28,29,31]. Chromosomal aberrations and unbalanced translocations were detected using the combinations of hTERT and SV40 as well as hTERT and HPV-E6/7 [28,29], whereas the karyotype of ADSCs immortalized by transducing with the combination of hTERT and murine Bmi-1 was normal [31]. Our established canine ADSC-K4DT and ADSC-K4D cells showed no change in chromosome number in 50 mitotic cells or chromosome aberrations in 20 mitotic cells. A previous study with detailed chromosome analysis of immortalized human fibroblasts transduced with K4DT showed that the incidence of chromosome abnormalities in the K4DT cells was similar to that in wild-type cells [32]. Therefore, cellular immortalization with K4DT is more advantageous by maintaining the original condition of chromosomes compared with oncogenic immortalization methods. Neither the ADSC-K4DT nor ADSC-K4D cells formed tumors at 16 weeks after subcutaneous injection into nude mice, indicating that these cells can be applied to transplantation therapy in vivo.

Except for CD90, the cell surface marker expression in the ADSC-K4DT and ADSC-K4D cells was similar to that in the primary ADSCs. CD90 expression was decreased as the cells proliferated, especially in the ADSC-K4DT cells. CD90 is highly expressed in MSCs regardless of the tissue source [33]. The detailed function of CD90 in MSCs remains unclear, but several studies have suggested that CD90 functions in MSC self-renewal, differentiation, and immunosuppression [34,35,36]. A study of MSCs isolated from various tissue sources, including adipose tissue with CD90 knockdown by a CD90-targeted small hairpin RNA lentiviral vector, showed that the reduction in CD90 expression did not affect the maintenance of MSC morphology, colony-forming ability, or proliferation [35]. However, CD90 knockdown MSCs had reduced CD44 and CD166 expression that enhanced osteogenic and adipogenic differentiation. Therefore, the authors suggested that a reduction in CD90 expression indicates a shift in the stemness state of MSCs towards a state more susceptible to differentiation [35]. We did not examine the expression of CD166, but highly expressed CD44 was maintained even when CD90 expression was reduced in the ADSC-K4DT and ADSC-K4D cells. There is no consensus on whether the characteristic immunosuppressive function of MSCs is affected by decreased CD90 expression [35,36]. A study cocultured human MSCs and PBMCs stimulated with PHA and then measured CD90 expression [36], which showed a negative correlation between CD90 expression and the lymphoproliferative response to PHA activation. Thus, it was suggested that human MSCs with suppressed CD90 expression had a reduced immunosuppressive effect by affecting T cell proliferation. However, another study also cocultured human CD90 knockdown MSCs and CFSE-labeled PBMCs stimulated with PHA and then analyzed lymphocyte proliferation [35], which showed no differences in lymphocyte proliferation between MSCs cocultured with CD90 knockdown or control MSCs. Thus, a reduction in CD90 expression did not affect the immunosuppressive effect of MSCs on lymphocyte proliferation. In our study, the inhibitory effect of the ADSC-K4DT and ADSC-K4D cells on PBMCs was not reduced compared with that of the primary ADSCs. Moreover, as CD90 expression decreased with increasing PDL, the inhibitory effect remained in the ADSC-K4DT and ADSC-K4D cells.

The pathway analysis revealed many pathway changes, especially in those related to the cell cycle and telomeres in the ADSC-K4DT and ADSC-K4D cells compared with the primary ADSCs. The changes in the differentially expressed genes in the cell cycle and cellular senescence pathways were similar between the ADSC-K4DT and ADSC-K4D cells compared with the primary ADSCs, suggesting that canine TERT transduction did not have a significant effect on cellular immortalization. However, further studies are needed to determine how the differences in gene expression and pathways in ADSC-K4DT and ADSC-K4D cells affect the properties and functions of ADSCs and to investigate the therapeutic effects of ADSC-K4DT and ADSC-K4D cells using in vivo experiments.

## 4. Materials and Methods

### 4.1. Animals

Three beagles (males; mean age: 1.5 years; mean body weight: 10.3 kg) were used to isolate ADSCs and PBMCs. Twenty-five male nude mice (5–6 weeks old) were used for the in vivo tumorigenic assays. The mice were housed in a temperature- and light-controlled room (12-h light/dark cycle) and had free access to water and standard laboratory food.

### 4.2. Isolation and Expansion of Canine Adipose-Derived Mesenchymal Stem Cells

Canine primary ADSCs were isolated and expanded as described previously [37]. In brief, after collecting falciform ligament adipose tissue from the anaesthetized dogs, the cells were digested with collagenase type I (Sigma-Aldrich, St. Louis, MO, USA), filtered through a 100-µm nylon mesh, and cultured in Dulbecco’s modified Eagle’s medium (DMEM) supplemented with 10% fetal bovine serum (FBS; Capricorn, Hessen, Germany) and a 1% antibiotic–antimycotic solution (Thermo Fisher Scientific, Waltham, MA, USA) at 37 °C in a humidified incubator containing 5% CO_2_. At 80–90% confluence, the cells were detached with a trypsin–EDTA solution (Sigma-Aldrich) and passaged repeatedly.

### 4.3. Construction of Lentiviral Vectors and Transduction of Canine ADSCs

Canine TERT, canine CCND1, and human CDK4R24C were used as immortalization-inducing genes. Lentiviral vectors were used to overexpress canine TERT, human CDK4R24C, and canine CCND1, namely pLV[Exp]-EGFP:T2A:Puro-EF1A > dgTERT and pLV[Exp]-Bsd-EF1A > dgCCND1[NM_001005757.1](ns):T2A:hCDK4R24C, which were constructed and packaged with VectorBuilder. The vector IDs VB900123-8680tqy and VB211026-1098frg can be used to retrieve detailed information about the vectors on vectorbuilder.com. The passage 3 ADSCs were seeded in 6-well plates and infected for 24 h in 1 mL DMEM with 4 µg/mL polybrene and lentivirus at MOI 50. At 48 h post-transduction, ADSCs were selected using 1 µg/mL puromycin and/or 6 µg/mL of blasticidin S for 7 days. Three types of transduced ADSCs were generated: (i) cells transduced with canine TERT (ADSC-TERT); (ii) cells transduced with human CDK4R24C, canine CCND1, and canine TERT (ADSC-K4DT); and (iii) cells transduced with human CDK4R24C and canine CCND1 (ADSC-K4D).

### 4.4. Population Doubling Analysis

To measure the proliferation rate in the long-term culture, the cells were evaluated by population doubling (PD) with sequential passages. The PD value was calculated with the following formula: PD = log2 (a/b), where “a” is the total number of cells recovered at the end of each passage, and “b” is the number of cells seeded at the start of each passage [30]. The PD level was the sum of the PD values obtained from each passage. The cells were cultured continuously by passaging for approximately 3 months.

### 4.5. Cell Cycle Analysis

The cell cycle analysis of the primary ADSCs (n = 3), ADSC-K4DT (n = 3) cells, and ADSC-K4D (n = 3) cells was performed and repeated in three independent experiments with flow cytometry using Cell Cycle Assay Solution Deep Red (DOJINDO, Kumamoto, Japan) in accordance with the manufacturer’s instructions.

### 4.6. Cell Surface Markers Analysis

The cell surface markers were analyzed with flow cytometry. The primary ADSCs at passage 3 (n = 3), and the ADSC-K4DT (n = 3) and ADSC-K4D (n = 3) cells were analyzed at low (approximately PDL 20), intermediate (approximately PDL 50), and high (approximately PDL 100) PDs. The cells were washed with FACS buffer (PBS containing 2% FBS), and the Fc receptors were blocked with canine Fc receptor binding inhibitor (Thermo Fisher Scientific). Then, the cells were incubated with the following phycoerythrin (PE)-conjugated antibodies: anti-CD29-PE (clone: TS2/16; BioLegend, San Diego, CA, USA), anti-CD34-PE (clone: 1H6; R&D Systems, Minneapolis, MN, USA), anti-CD44-PE (clone: IM7; BioLegend), anti-CD45-PE (clone: YKIX716.13; eBioscience, San Diego, CA, USA), anti-CD90-PE (clone: YKIX337.217: eBioscience), and anti-HLA-DR-PE (clone: G46-6: BD Biosciences, Franklin Lake, NJ, USA). All the cells at each passage were examined in triplicate.

### 4.7. Trilineage Differentiation Assay

For adipogenic differentiation, the cells were seeded on 12-well plates (4 × 10^4^ cells/well) and cultured in DMEM supplemented with 10% FBS and a 1% antibiotic–antimycotic solution until confluency. Then, the medium was changed to StemPro Adipogenesis Differentiation Kit (Thermo Fisher Scientific). The medium was changed twice weekly. Adipogenesis was analyzed with oil red O staining after 28 days.

For osteogenic differentiation, the cells were seeded on 12-well plates (2 × 10^5^ cells/well) and cultured in DMEM supplemented with 10% FBS and a 1% antibiotic–antimycotic solution. The next day, the medium was changed to StemPro Osteogenesis Differentiation Kit (Thermo Fisher Scientific). The medium was changed twice weekly. Osteogenesis was analyzed with Alizarin red staining after 21 days.

For chondrogenic differentiation, a MesenCult-ACF Chondrogenic Differentiation Kit (STEMCELL Technologies, Vancouver, BC, Canada) was used. In brief, 1 × 10^6^ cells suspended in 0.5 mL medium were placed in a 15 mL polypropylene tube and centrifuged at 300× *g* for 5 min. Then, the tube cap was loosened, and the cells were incubated at 37 °C with 5% CO_2_. The medium was changed every 3 days. After 21 days, the cell pellet was fixed in 10% formalin and embedded in paraffin. After fixation, the cell pellet was cut into 6-µm-thick sections and stained with Alcian blue.

### 4.8. Cellular Senescence Staining

The cells were seeded in a six-well plate (8 × 10^4^ cells/well). After 24 h, β-galactosidase expression was detected using a SA-β-gal staining Kit (Cell Signaling Technology, Danvers, MA, USA) in accordance with the manufacturer’s instructions.

### 4.9. Polymerase Chain Reaction

Total RNA was extracted from the cultured cells using a NucleoSpin RNA kit (TaKaRa, Shiga, Japan) in accordance with the manufacturer’s instructions. cDNA was synthesized from 0.5 µg total RNA in accordance with the manufacturer’s instructions. The PCR amplification was performed using GoTaq Green Master Mix (Promega, Madison, WI, USA) based on the protocol provided by the manufacturer. The primer sequences were as follows: Canine CCND1: forward, 5′- GGTCTGCGAGGAGCAGAAGT-3′ and reverse, 5′- GATGAAGTCGTGTGGGGTCA-3′ (286-bp product size); canine TERT: forward, 5′- TTTGCAGACCTGCAGCCTTA-3′ and reverse, 5′- CACTGGCTGGTTGAATGGAA-3′ (780-bp product size); and human CDK4R24C: forward, 5′- AGTGGCTGAAATTGGTGTCG-3′ and reverse, 5′- ATGTGGCACAGACGTCCATC-3′ (218-bp product size). The PCR products were separated with 2% agarose gel electrophoresis and stained with Midori Green Xtra (NIPPON Genetics, Tokyo, Japan).

### 4.10. Karyotype Analysis

Karyotype analysis was carried out in the primary ADSCs, and the ADSC-K4DT and ADSC-K4D cells by Nihon Gene Research Laboratories Inc. In brief, the cells were treated with colcemid overnight to increase the number of cells in metaphase. After trypsinization, the cells were treated with a hypotonic solution, fixed, stained with a Giemsa solution, and analyzed for detailed chromosomal patterns with G-banding.

### 4.11. In Vivo Tumorigenic Assay

Nude mice were used to assess the tumorigenicity of the primary ADSCs, and the ADSC-K4DT and ADSC-K4D cells. The cells suspended in PBS were implanted in nude mice by subcutaneous injection. HeLa cells were used as a positive control. Twenty-five nude mice were equally divided into five groups. Group 1 was subcutaneously injected with primary ADSCs (left trunk) and Hela cells (right trunk). Group 2 was subcutaneously injected with ADSC-K4DT cells (left) and Hela cells (right). Group 3 was subcutaneously injected with ADSC-K4D cells (left) and Hela cells (right). Group 4 was subcutaneously injected with primary ADSCs (left) and ADSC-K4DT cells (right). Group 5 was subcutaneously injected with ADSC-K4D cells (left) and 0.1 mL PBS (right). A total of 1 × 10^6^ cells was implanted at each injection site. Groups 1–3 were euthanized with CO_2_ asphyxiation at 30 days after implantation. Groups 4 and 5 were euthanized with CO_2_ asphyxiation at 16 weeks after implantation. Tumor nodules were removed and fixed in 4% paraformaldehyde. After fixation and embedding in paraffin, the tissues were cut into 4-µm-thick sections and stained with hematoxylin and eosin.

### 4.12. Lymphocyte Proliferation Assay

The primary ADSCs, and the ADSC-K4DT and ADSC-K4D cells (2 × 10^5^ cells/well) were seeded in flat-bottomed 24-well plates in RPMI 1640 medium supplemented with 10% FBS and a 1% antibiotic–antimycotic solution. Canine blood was collected from the jugular vein of healthy beagles into heparinized tubes. The PBMCs were immediately isolated with density-gradient centrifugation using Histopaque-1077 (Sigma-Aldrich) and SepMate-15 (VERITAS, Tokyo, Japan). The PBMCs were prelabeled with a 5-µM CFSE solution using a CFSE Cell Division Tracer Kit (BioLegend) before seeding in accordance with the manufacturer’s instructions, and 1 × 10^6^ PBMCs were added to the wells with or without primary ADSCs, and ADSC-K4DT and ADSC-K4D cells. To evaluate the lymphocyte proliferation in the presence of primary ADSCs, and ADSC-K4DT and ADSC-K4D cells, the lymphocytes were activated with 5 µg/mL ConA and maintained at 37 °C with 5% CO_2_ for 3 days. After coculture, the PBMCs were collected and washed with FACS buffer, and the PBMC proliferation was measured with flow cytometry.

### 4.13. RNA-Sequencing

We performed RNA-seq of total RNA samples isolated from the primary ADSCs, and high PDs (approximately PDL 50) of the ADSC-K4DT and ADSC-K4D cells. The cDNA library construction was carried out with 1 µg total RNA using a NEBNext Ultra II RNA Library Prep Kit from Illumina in accordance with the manufacturer’s instructions followed by paired-end sequencing (2 × 150 bp) using a Novaseq6000. For each library, an average of 16–20 million read pairs were generated. Quality control checks of the sequencing raw data were conducted with FastQC ver 0.23.2. Adapter trimming was performed with Trim Galore. The relative expression of the transcripts was quantified in each sample using featureCount (ver 2.0.1). The Fastq files were mapped to the reference genome for *Canis lupus familiaris* using STAR software ver 2.7.10a. The differentially expressed genes (upregulated or downregulated genes) were determined using edgeR ver 3.22.2 with an adjusted *p*-value of < 0.05 and fold change of >2 or <0.5. Gene set enrichment analysis was performed with RaNa-seq (https://ranaseq.eu/ accessed on 28 September 2022).

### 4.14. Statistical Analysis

All data are presented as the mean ± standard deviation. The differences among multiple groups were assessed with one- or two-way analysis of variance. The differences were compared using the Tukey–Kramer post-hoc test. *p* < 0.05 was considered statistically significant. The statistical analyses were performed using Excel 2019 with add-in software Statcel 3.

## 5. Conclusions

We established two types of immortalized canine ADSCs, ADSC-K4DT and ADSC-K4D cells, by transduction with combinations of human CDK4R24C, canine CCND1, and canine TERT. These cells had high proliferative and anti-senescence abilities in addition to the fundamental characteristics of primary ADSCs except for CD90 expression. Although further functional analysis is needed, ADSC-K4DT and ADSC-K4D cells have the potential to be a novel cell source for stem cell therapy, which may reduce the number of donors and achieve stable therapeutic effects.

## Figures and Tables

**Figure 1 ijms-24-02250-f001:**
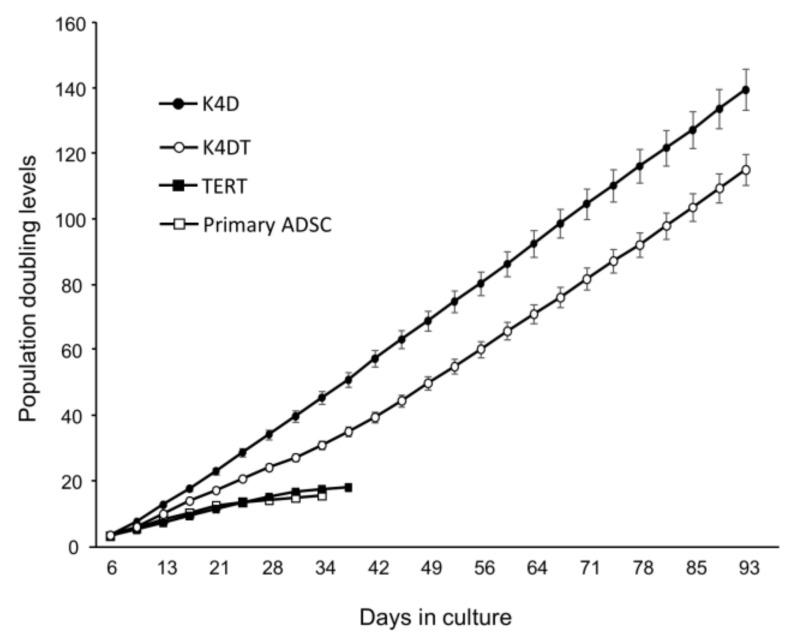
Growth curves of the primary ADSCs, and the ADSC-TERT, ADSC-K4DT, and ADSC- K4D cells. Population doubling levels (PDLs) were calculated for each cell line over time. PDL represents the sum of the population doubling (PD) values obtained when the cells were passaged. The histogram represents the mean of the three independent experiments. Data are expressed as the mean ± standard deviation.

**Figure 2 ijms-24-02250-f002:**
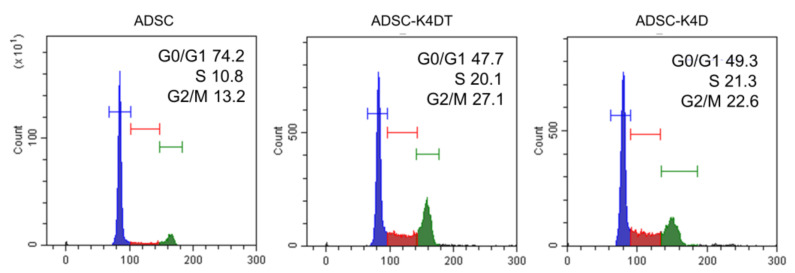
Cell cycle analysis of the primary ADSCs, and the ADSC-K4DT and ADSC-K4D cells.

**Figure 3 ijms-24-02250-f003:**
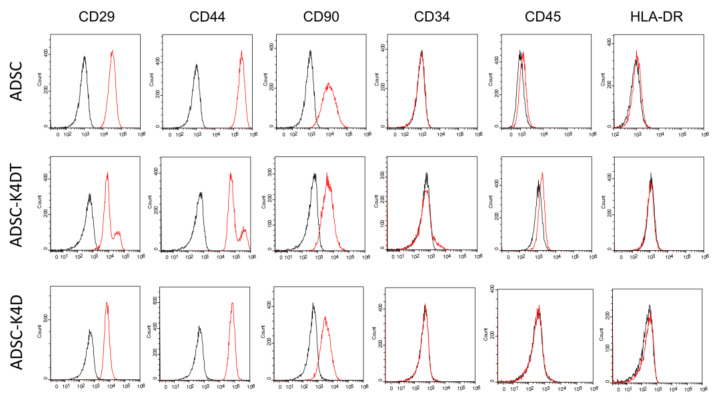
Expression of cell surface markers analyzed with flow cytometry. The primary ADSCs at passage 3, and the ADSC-K4DT and ADSC-K4D cells after antibiotic selection are shown. Black lines represent isotype controls, and red lines indicate each cell line.

**Figure 4 ijms-24-02250-f004:**
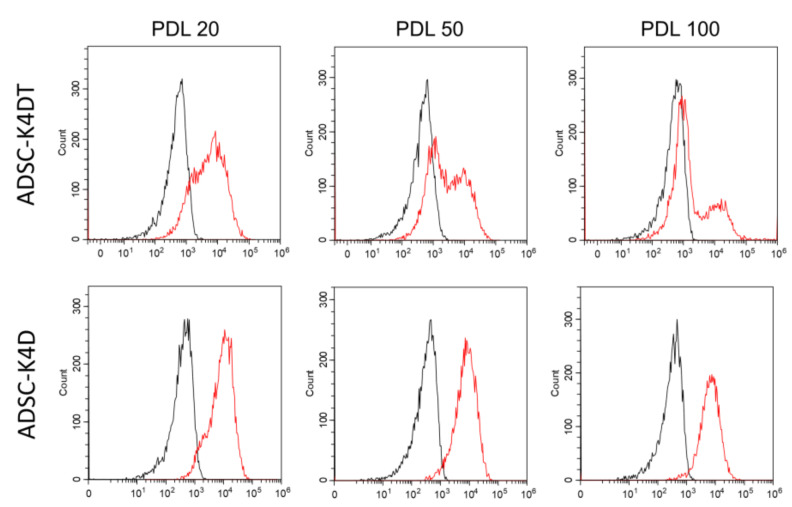
Changes in CD90 expression in the ADSC-K4DT and ADSC-K4D cells. CD90 expression in both cell lines at low (approximately PDL 20), intermediate (approximately PDL 50), and high (approximately PDL 100) PDs is shown. Black lines represent isotype controls, and red lines indicate each cell line. CD90 expression in the ADSC-K4DT cells decreased gradually in accordance with cell division, but that in the ADSC-K4D cells was maintained after repeated cell division.

**Figure 5 ijms-24-02250-f005:**
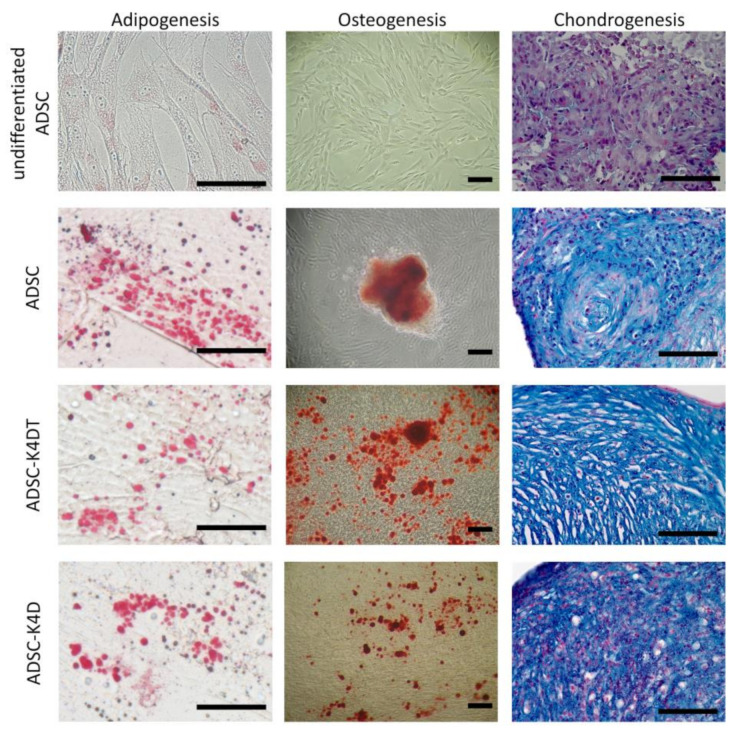
Trilineage differentiation potential of the ADSC-K4DT and ADSC-K4D cells. Adipocytes, osteocytes, and chondrocytes were induced from the ADSCs at passage 3, and the ADSC-K4DT and ADSC-K4D cells at approximately PDL 100 by each differentiation culture method and stained with oil red O, alizarin red, and Alcian blue, respectively. Bar = 100 µm.

**Figure 6 ijms-24-02250-f006:**
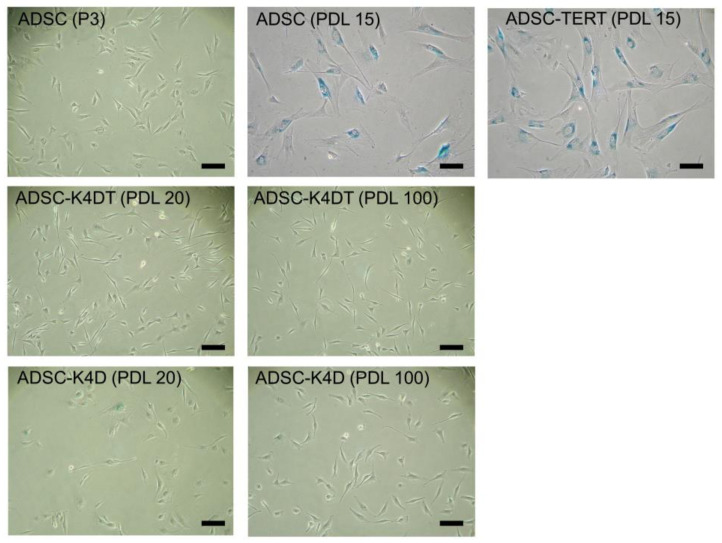
SA-β-gal expression in the primary ADSCs and ADSCs post-transduction. Senescence cells were not detected among the primary ADSCs at passage 3, but most of the primary ADSCs at PDL 15 (passage 7) were stained with SA-β-gal. The ADSC-TERT cells at PDL 15 had almost ceased proliferation and were identified as senescence cells similarly to the primary ADSCs at PDL 15. Few senescence cells were detected with SA-β-gal staining among the ADSC-K4DT and ADSC-K4D cells even at PDL 100. Bar = 100 µm.

**Figure 7 ijms-24-02250-f007:**
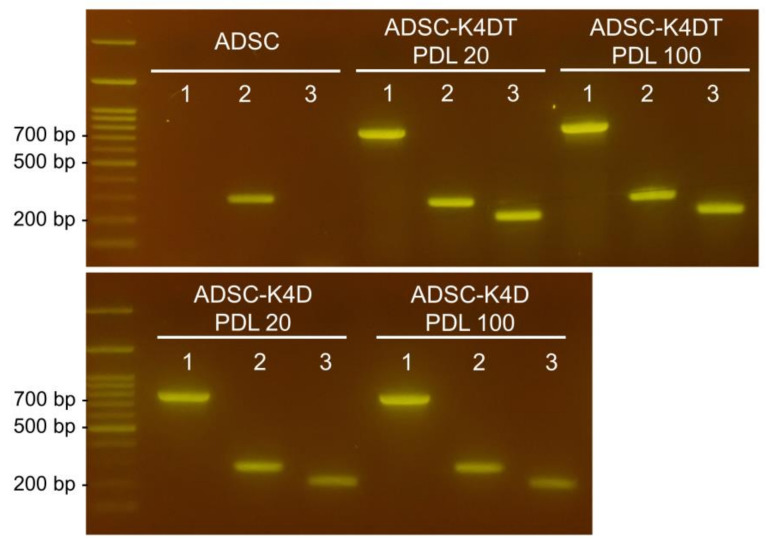
Detection of transduced genes with PCR. Lane 1: canine TERT; Lane 2: canine CCND1; Lane 3: human CDK4R24C.

**Figure 8 ijms-24-02250-f008:**
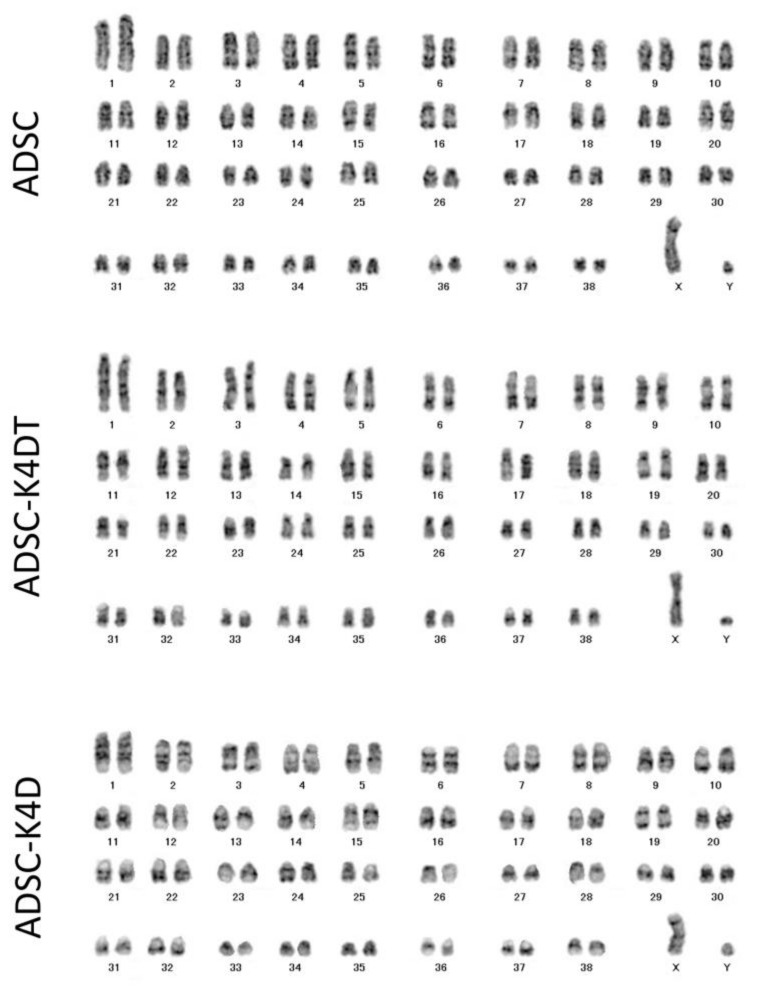
Karyotype analyses of the ADSC-K4DT and ADSC-K4D cells. Aligned chromosomes of the primary ADSCs, and the ADSC-K4DT and ADSC-K4D cells are shown. All cells had a 2 n = 76 + XY pattern.

**Figure 9 ijms-24-02250-f009:**
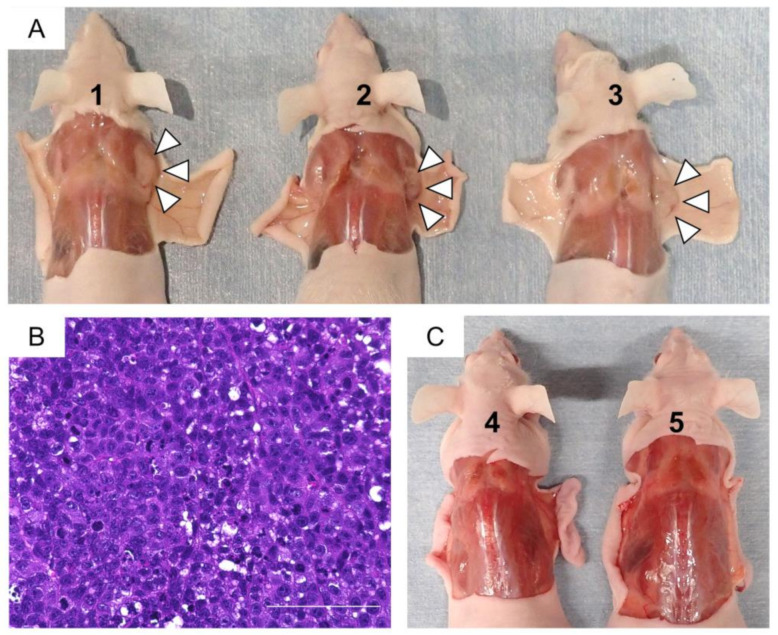
Tumorigenicity of the ADSC-K4DT and ADSC-K4D cells. (**A**) Inoculated Hela cells developed masses at injection sites (right side of the nude mice indicated by arrowheads), but no masses were generated by the ADSCs (No. 1), ADSC-K4DT cells (No. 2), or ADSC-K4D cells (No. 3) at the injected sites (left side of the nude mice) after 30 days. (**B**) Hematoxylin–eosin staining of tumor nodules derived from Hela cell-injected sites (scale bar = 100 µm). (**C**) After 16 weeks, there were no generated masses in group 4 (left; primary ADSCs, right; ADSC-K4D cells, No. 4) or group 5 (left; ADSC-K4DT cells, right; PBS, No. 5).

**Figure 10 ijms-24-02250-f010:**
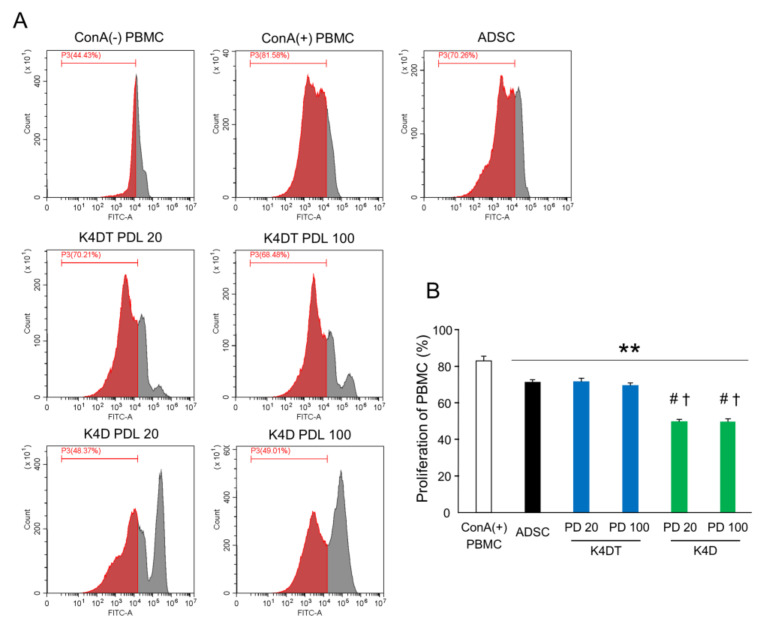
Proliferative ability of PBMCs cocultured with ADSC-K4DT and ADSC-K4D cells. (**A**) PBMCs were stained with carboxyfluorescein succinimidyl ester (CFSE) and cultured for 72 h with primary ADSCs, ADSC-K4DT, or ADSC-K4D cells. The PBMC proliferation was quantified as the percentage of CFSE low cells. (**B**) Comparison of proliferative PBMCs stimulated with ConA when cocultured with ADSC-K4DT and ADSC-K4D cells. The inhibitive effect on the PBMCs cocultured with primary ADSCs, and ADSC-K4DT and ADSC-K4D cells was significantly enhanced compared to only stimulation with ConA. The inhibitive effect on the PBMCs was not different between PDL 20 and PDL 100 of the ADSC-K4DT and ADSC-K4D cells, respectively. Data are expressed as the mean ± standard deviation; n = 6; ** *p* < 0.01 vs. ConA (+) PBMCs, ^#^
*p* < 0.05 vs. ADSCs, ^†^
*p*< 0.05 vs. ADSC-K4DT cells.

**Figure 11 ijms-24-02250-f011:**
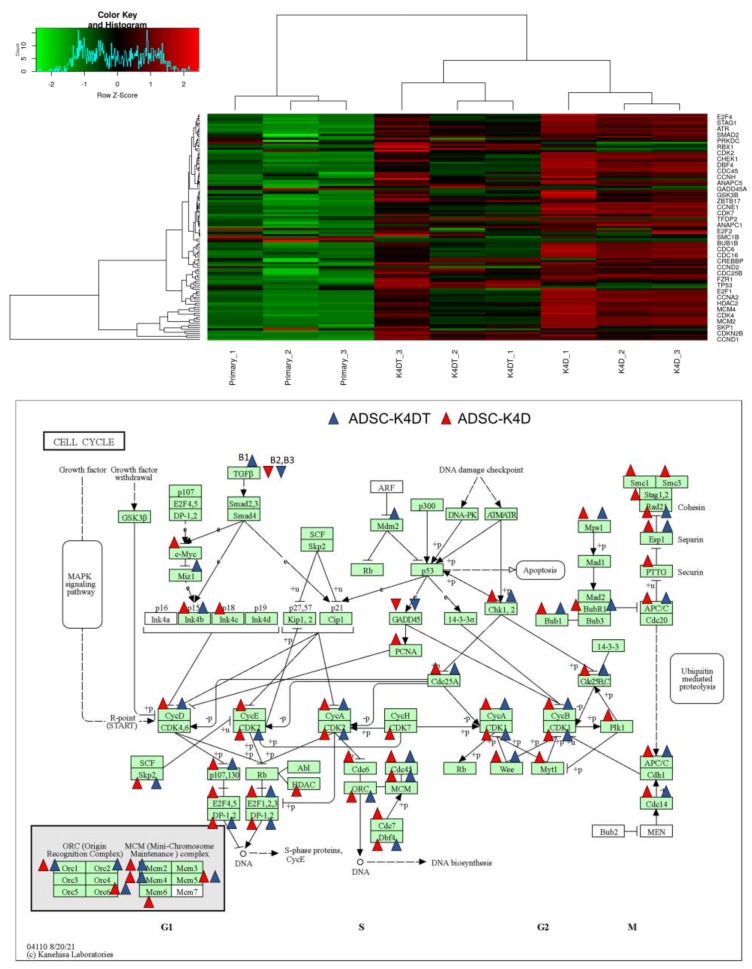
RNA-seq analysis of the cell cycle-related genes. Upper panel: Heatmap of the genes listed in the cell cycle pathways in the primary ADSCs, and the ADSC-K4DT and ADSC-K4D cells. Red indicates high expression, and green indicates low expression. Lower panel: Summary of the up- and down-regulated genes among the KEGG cell cycle-related genes. Arrowheads indicate the up- or down-regulated genes in the ADSC-K4DT and ADSC-K4D cells. The genes with an adjusted *p*-value of < 0.05 and fold change of >2 or <0.5 compared with the primary ADSCs were mapped.

**Figure 12 ijms-24-02250-f012:**
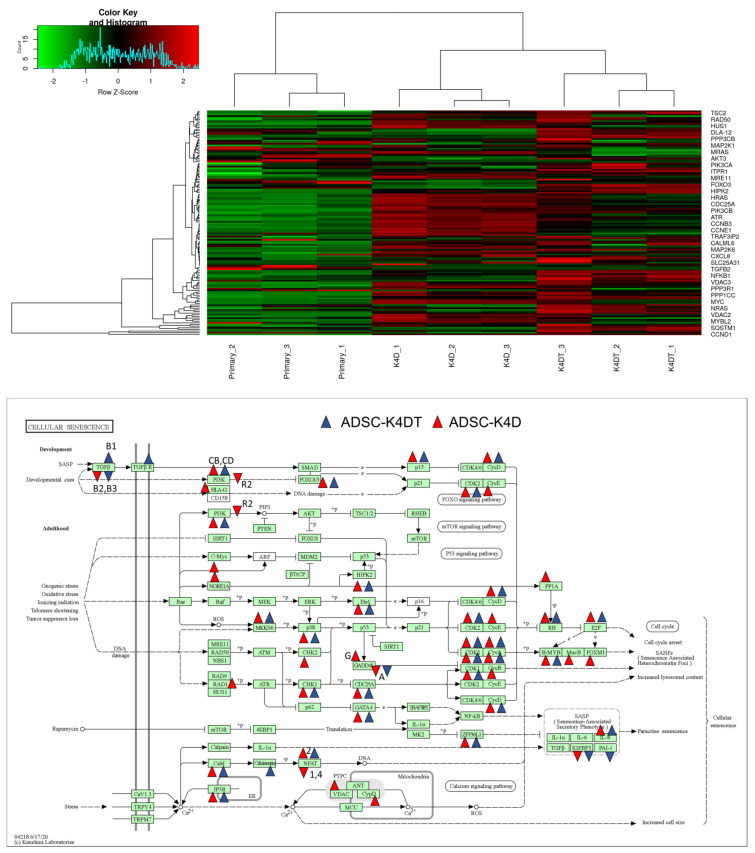
RNA-seq analysis of the cell senescence-related genes. Upper panel: Heatmap of the genes listed in the cell senescence pathways in the primary ADSCs, and the ADSC-K4DT and ADSC-K4D cells. Red indicates high expression, and green indicates low expression. Lower panel: Summary of the up- and down-regulated genes among the KEGG cell senescence-related genes. Arrowheads indicate the up- or down-regulated genes in the ADSC-K4DT and ADSC-K4D cells. The genes with an adjusted *p*-value of < 0.05 and fold change of >2 or <0.5 compared with the primary ADSCs were mapped.

**Table 1 ijms-24-02250-t001:** Flow cytometric analysis of cell surface markers.

		CD29	CD44	CD90	CD34	CD45	HLA-DR
ADSC	passage 3	97.6 ± 1.8	98.7 ± 1.4	97.2 ± 1.5	0.4 ± 0.1	0.4 ± 0.2	0.4 ± 0.2
ADSC-K4DT	PDL 20	96.4 ± 1.4	99.1 ± 1.3	67.2 ± 2.1 *	0.5 ± 0.3	0.4 ± 0.2	0.4 ± 0.3
	PDL 50	97.5 ± 1.1	97.3 ± 1.4	37.5 ± 3.2 **^,#^	0.9 ± 0.3	0.4 ± 0.2	0.5 ± 0.3
	PDL 100	98.3 ± 1.3	97.4 ± 1.1	19.8 ± 3.3 **^,#^	0.4 ± 0.1	0.4 ± 0.2	0.4 ± 0.4
ADSC-K4D	PDL 20	98.0 ± 1.3	98.4 ± 1.5	88.0 ± 4.1 *	0.5 ± 0.2	0.4 ± 0.2	0.4 ± 0.2
	PDL 50	98.1 ± 1.5	98.1 ± 1.3	89.3 ± 1.8 *	0.5 ± 0.2	0.4 ± 0.2	0.5 ± 0.4
	PDL 100	98.6 ± 1.6	98.5 ± 1.0	90.0 ± 2.4 *	0.4 ± 0.3	0.4 ± 0.2	0.4 ± 0.2

Data are expressed as the percentage of positive cells (mean ± standard deviation); *n* = 3 and three independent experiments; * *p* < 0.05 vs. ADSCs, ** *p* < 0.01 vs. ADSCs, ^#^
*p* < 0.05 vs. ADSC-K4DT cells at PDL 20.

## Data Availability

The data supporting the findings of this study are available from the corresponding author upon reasonable request.

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
