# Peer review of "Immortalized Canine Adipose-Derived Mesenchymal Stem Cells as a Novel Candidate Cell Source for Mesenchymal Stem Cell Therapy"

_ijms, 2023, doi:10.3390/ijms24032250_

Round 1

Reviewer 1 Report

The efficacy of several MSC clinical trials was not as good as the pre-clinical findings. The variations of heterogenic MSCs may be the critical reason that limited its clinical application. Yasumura et al. aimed to set up immortalized adipose-derived MSCs as a candidate cell of stable source of MSC. This attempt is commendable for the MSC therapy field. However, several critical concerns should be addressed before resubmission.

Major concerns:

1. The use of canine adipose but not human tissue significantly affects the importance of this study. 

2. This study aims to generate MSC for MSC therapy. However, no in vivo experiment was performed for the therapeutic effect of immortalized MSC.

3. Genomic modification of immortalized MSCs inevitably causes safety concerns for clinical trials. If this study aims to generate a new candidate for MSC therapy, how to balance the safety concerns and the stabilized therapeutic effects?

4. There are also lots of flaws in data presentation that the authors should address.

Detailed concerns about the figures: 

  1. Figure 1, the Standard Deviation of the histogram should be given. 
  2. Figure 2, how many independent experiments have been performed for cell cycle? N should be given. Why the author chose PDL 50 to compare the difference between primary and informalized cells? Why not compare them at the same PDL level, for example, PDL 15?
  3. Figures 3, and 4, statistical results should be given for the Flow data.
  4. Table 1, repetitive ADSC-K4DT groups. Why not list PDL 50, 100, and 20 in order?
  5. Figure 5, it is not enough to just use staining to verify the multiple differentiation capacity of MSC.
  6. Figure 6, it is not enough to just use SA-β-gal to evaluate senescence cells.
  7. Figure 9A. It is confusing what cells were inoculated to these three nude mice. Does the figure legend indicate #1, 2, 3 are both ADSCs and Hela cells?

Author Response

Response to Reviewer 1 Comments

Point 1: The use of canine adipose but not human tissue significantly affects the importance of this study.

Response 1: The current study is aimed at the therapeutic application of mesenchymal stem cells in veterinary clinical practice, especially in dogs. Therefore, we have analyzed immortalized canine adipose tissue-derived mesenchymal stem cells.

Point 2: This study aims to generate MSC for MSC therapy. However, no in vivo experiment was performed for the therapeutic effect of immortalized MSC.

Response 2: We also believe it is important to evaluate the therapeutic effect of immortalized canine ADSCs in animal models or clinical cases. In this study, we have established the immortalized canine ADSCs by transduction with combinations of CDKR24C, cyclin D1, and TERT, and analyzed the biological characteristics of these cells in vitro. We are planning in vivo study using animal models in a further study after accepted in International Journal of Molecular Sciences. Therefore, we have added the description: “..., and to investigate the therapeutic effects of ADSC-K4DT and ADSC-K4D cells in vivo experiments.” in the last of Discussion.

Point 3: Genomic modification of immortalized MSCs inevitably causes safety concerns for clinical trials. If this study aims to generate a new candidate for MSC therapy, how to balance the safety concerns and the stabilized therapeutic effects?

Response 3: In this study, we have confirmed maintenance of the chromosome conditions after transduced genes and non-tumorigenicity in mice. However, as you point out, only these results do not indicate that established canine immortalized ADSCs (ADSC-K4DT and ADSC-K4D cells) are truly safe for clinical use. In addition to further study using animal models (above Point 2), we are planning to research cell-free therapy using secretomes, such as microvesicles and exosomes, from immortalized ADSCs for clinical therapy.

Point 4: There are also lots of flaws in data presentation that the authors should address.

Response 4: We have corrected data presentation as your following advice.

Point 5: Figure 1, the Standard Deviation of the histogram should be given.

Response 5: We have revised the Figure 1.

Point 6: Figure 2, how many independent experiments have been performed for cell cycle? N should be given. Why the author chose PDL 50 to compare the difference between primary and informalized cells? Why not compare them at the same PDL level, for example, PDL 15?

Response 6: We have added the number of cell lines and experiments in “4.5. Cell cycle analysis.” The reason why we have chosen PDL 50 to compare cell cycle analysis between primary ADSC and immortalized cells is that to analyze cells that have been exactly established as immortalized ADSC cells. Since the PDL of primary ADSCs was approximately 15-20, we have examined immortalized cells at PDL 50 which was considered to ensure that immortalization has been established.

Point 7: Figures 3, and 4, statistical results should be given for the Flow data.

Response 7: We apologized for confusing Figure 3 and 4 and Table 1. We have showed the statistical results of expression of surface markers in Table 1. We have changed the expression of cell surface markers in primary ADSCs from at passage 2 to passage 3, and the percentage of positive cells in ADSCs, ADSC-K4DT, and ADSC-K4D cells have been shown in Table 1.

Point 8: Table 1, repetitive ADSC-K4DT groups. Why not list PDL 50, 100, and 20 in order?

Response 8: We apologized for confusing in Table 1. In original Table 1, we have listed PDL 20, 50, and 100 in order. We have revised Table 1 by adding a line to make the order clearly.

Point 9: Figure 5, it is not enough to just use staining to verify the multiple differentiation capacity of MSC.

Response 9: The purpose of confirming trilineage differentiation (adipocyte, osteocyte, and chondrocyte) was to determine whether the immortalized ADSC cells maintained the minimal criteria for defining mesenchymal stem cells. International Society for Cellular Therapy (ISCT) propose three criteria to define MSC: 1) adherence to plastic, 2) specific surface antigen expression, and 3) multipotent differentiation potential (In vitro differentiation: osteoblasts, adipocytes, chondroblasts demonstrated by staining of in vitro cell culture). Therefore, we have examined the trilineage differentiation capacity by staining of in vitro cell culture.

Point 10: Figure 6, it is not enough to just use SA-β-gal to evaluate senescence cells.

Response 10: We have added the description of morphological changes related to cellular senescence referring to research of immortalized cells (Reference #31) as follow: “Primary ADSCs and ADSC-TERT cells showed enlarged cytoplasm at approximately PDL 15, but the ADSC-K4DT and ADSC-K4D cells showed no morphological changes (Figure 6).” in 2.5. Lack of cellular senescence in ADSC-K4D and ADSC-K4DT cells

Point 11: Figure 9A. It is confusing what cells were inoculated to these three nude mice. Does the figure legend indicate #1, 2, 3 are both ADSCs and Hela cells?

Response 11: ADSCs were injected at left side, and Hela cells were injected at right side of #1, 2, and 3 mice. We have revised the description in Figure 9 legend as follows: Inoculated Hela cells developed masses at injection sites (right side of nude mice indicated by arrow-heads), but no masses were generated by ADSCs (No. 1), ADSC-K4DT cells (No. 2), or ADSC-K4D cells (No. 3) at injected sites (left side of nude mice) after 30 days.

Reviewer 2 Report

Why stem cells  seeding density  vary in  chondro- osteo- and adipogenic experiment? Figure 5 and 6   improve image resolution.  It is not clear  if only markers expression  was performed using cells from three donors and  in three replicates. Population doubling analysis – Authors should provide the references. Figure 5, the control group ( undifferentiated ADSC ) should be presented. Figure 5 and 6, the scale bar or  magnification  should be included.  Figure 6 the quality  is poor, use a better picture for adipo differentiation, with a  higher focus on the vesicles. Figure 10 A  improve  quality of  figure description. Figure 11 B and 12 B improve graph resolution.  In the current version the figures are illegible.  Why stem cells from different passages were used in individual experiments  (passage 3 - karyotype analysis of  primary ADSCs,   passage 2- flow cytometric analysis of cell surface markers).

Author Response

Response to Reviewer 2 Comments

Point 1: Why stem cells seeding density vary in chondro- osteo- and adipogenic experiment?

Response 1: We are not sure as to the definite reason for different cell seeding densities, but we have followed the manufacturer’s instructions to determine the seeding density.

Point 2: Figure 5 and 6 improve image resolution.

Response 2: We have changed all figures including Figure 5 and 6 to match “Instructions for Authors” (minimum 1000 pixels width/height, or a resolution of 300 dpi or higher).

Point 3: It is not clear if only markers expression was performed using cells from three donors and in three replicates.

Response 3: We have revised the manuscript clearly for noted using cells from three donors and in triplicate as follows: Cell surface markers were analyzed by flow cytometry. Primary ADSCs at passage 3 (n=3) and ADSC-K4DT (n=3) and ADSC-K4D (n=3) cells were analyzed at low (approxi-mately PDL 20), intermediate (approximately PDL 50), and high (approximately PDL 100) PDs.... All cells at each passage were examined in triplicate.

Point 4: Population doubling analysis – Authors should provide the references

Response 4: We have added the references #31 (Orimoto, A.; Kyakumoto, S.; Eitsuka, T.; Nakagawa, K.; Kiyono, T.; Fukuda, T. Efficient immortalization of human dental pulp stem cells with expression of cell cycle regulators with the intact chromosomal condition. PLoS One. 2020, 15, e0229996.)

Point 5: Figure 5, the control group (undifferentiated ADSC) should be presented.

Response 5: We have added the control group (undifferentiated ADSC) in Figure 5.

Point 6: Figure 5 and 6, the scale bar or magnification should be included.

Response 6: We have changed the Figure 5 and 6, and included the scale bar.

Point 7: Figure 6 the quality is poor, use a better picture for adipo differentiation, with a higher focus on the vesicles.

Response 7: We have changed the Figure 6 in which clearly identify fat droplets.

Point 8: Figure 10 A improve quality of figure description.

Response 8: We have revised the legend of Figure 10 A as follows: PBMCs were stained with carboxyfluorescein succinimidyl ester (CFSE) and cultured for 72 h with primary ADSCs, ADSC-K4DT, or ADSC-K4D cells. PBMC proliferation was quantified as the percentage of CFSE low cells.

Point 9: Figure 11B and 12B improve graph resolution. In the current version the figures are illegible.

Response 9: We have changed the Figure 11B and 12B to match “Instructions for Authors” (minimum 1000 pixels width/height, or a resolution of 300 dpi or higher).

Point 10: Why stem cells from different passages were used in individual experiments (passage 3 - karyotype analysis of primary ADSCs, passage 2- flow cytometric analysis of cell surface markers).

Response 10: We have changed the data in flow cytometric analysis of primary ADSCs from P2 to P3.

Reviewer 3 Report

Dear Authors 

I have evaluated your interesting manuscript entitled “Immortalized canine adipose-derived mesenchymal stem cells as a novel candidate cell source for mesenchymal stem cell therapy”, your work is well done but some corrections are needed to better present your research as below:

1.     The abbreviation list is needed before introduction

2.     As you are study the novel ADSCs to be grafted to cell therapy. In the introduction you should focus on this issue that why you are going to use the alternative candidate cell source instead of human cell source?  

3.     Your abstract is need to be quantitative not just present your brief research that carried out (your abstract is presented qualitative). For example: … showed a dramatic increase in proliferation…. (Percent???) and so on.

4.     As a question, did you have investigate the immunogenicity response of transduced ADSCs when grafted to mouse?

5.     I strongly encourage you to make a graphical abstract of your process.

Best wishes

Author Response

Response to Reviewer 3 Comments

Point 1: The abbreviation list is needed before introduction.

Response 1: The list of abbreviations was not mentioned in “Instructions for Authors.” If it is possible to include a list of abbreviations, please indicate so we can add it.

Point 2: As you are study the novel ADSCs to be grafted to cell therapy. In the introduction you should focus on this issue that why you are going to use the alternative candidate cell source instead of human cell source?

Response 2: The current study is aimed at the therapeutic application of mesenchymal stem cells in veterinary clinical practice, especially in dogs. Therefore, we have analyzed immortalized canine adipose tissue-derived mesenchymal stem cells. We have described the reason why attempted to establish the immortalized canine ADSCs in Introduction as follows: ... Therefore, donor-to-donor variability of the phenotype and growth kinetics causes significant inter-individual heterogeneity in the secreted factors of ADSCs. This potentially results in inconsistent outcomes in clinical trials and prevents the practical application of stem cell therapies [16,19]. A solution to avoid variations in ADSCs and stabilize the therapeutic effects is to restrict donors, but this is impossible because of the limited proliferative capacity of primary ADSCs. Therefore, in this study, we attempted to immortalize canine ADSCs.

Point 3: Your abstract is need to be quantitative not just present your brief research that carried out (your abstract is presented qualitative). For example: … showed a dramatic increase in proliferation…. (Percent???) and so on.

Response 3: We appreciate for your advice. We have added the PDL values in abstract. It is difficult to present quantitative data of our research in abstract (The abstract should be a total of about 200 words maximum. Revised abstract is 237 words). We added the graphic abstract as your advice.

Point 4: As a question, did you have investigate the immunogenicity response of transduced ADSCs when grafted to mouse?

Response 4: We have not investigated the immunogenicity response of transduced ADSCs when grafted to mouse. However, we also believe it is important to evaluate the therapeutic effect of immortalized canine ADSCs in animal models or clinical cases. In this study, we have established the immortalized canine ADSCs by transduction with combinations of CDKR24C, cyclin D1, and TERT, and analyzed the biological characteristics of these cells in vitro. We are planning in vivo study using animal models in a further study after accepted in International Journal of Molecular Sciences. Therefore, we have added the description: “..., and to investigate the therapeutic effects of ADSC-K4DT and ADSC-K4D cells in vivo experiments.” in the last of Discussion.

Point 5: I strongly encourage you to make a graphical abstract of your process.

Response 5: We have added a graphic abstract.

Round 2

Reviewer 1 Report

The authors have addressed most of my concerns. However, there are still some issues that should be addressed. 

1. In Table 1, it is confused that there are still repetitive ADSC-K4DT groups. 

2. In Figure 5, the authors used identical oil red staining images in different groups in the original version, which were replaced in this version. The authors should check their original data carefully.

Author Response

Point 1: In Table 1, it is confused that there are still repetitive ADSC-K4DT groups.

Response 1: We apologize for the lack of confirmation. We have corrected lower line to ADSC-K4D.

Point 2: In Figure 5, the authors used identical oil red staining images in different groups in the original version, which were replaced in this version. The authors should check their original data carefully.

Response 2: We have used the original version of the photos in the current revised manuscript to able to confirm fat droplets clearly.